# Organisation and delivery of a dedicated multidisciplinary prone ventilation team in the intensive care unit: Strategies and lessons from COVID-19

Luke Bracegirdle[1,2,3,4]*, Matthew Stubbs[1], Rezaur Rahman[1,4], Alexander I. R. Jackson[1,2,3,4], Helmi C. Burton-Papp[4], Robert Chambers[1,4], Sanjay Gupta[1,4], Michael P. W. Grocott[1,2,3,4], Ahilanandan Dushianthan[2,3,4]

1 Shackleton Department of Anaesthesia, University Hospital Southampton NHS Foundation Trust, Southampton, United Kingdom, 2 Southampton NIHR Biomedical Research Centre, University Hospital Southampton NHS Foundation Trust, Southampton, United Kingdom, 3 Faculty of Medicine, University of Southampton, University Hospital Southampton, Southampton, United Kingdom, 4 General Intensive Care Unit, University Hospital Southampton NHS Foundation Trust, Southampton, United Kingdom

* luke.bracegirdle@doctors.org.uk

**Data Availability Statement:** All relevant data are within the paper and supporting files.

## Abstract

### Background

COVID-19 placed immense strain on healthcare systems, necessitating innovative responses to the surge of critically ill patients, particularly those requiring mechanical ventilation. In this report, we detail the establishment of a dedicated critical care prone positioning team at University Hospital Southampton in response to escalating demand for prone positioning during the initial wave of the pandemic.

### Methods

The formation of a prone positioning team involved meticulous planning and collaboration across disciplines to ensure safe and efficient manoeuvrers. A comprehensive training strategy, aligned with national guidelines, was implemented for approximately 550 staff members from a diverse background. We surveyed team members to gain insight to the lived experience.

### Results

A total of 78 full-time team members were recruited and successfully executed over 1200 manoeuvres over an eight-week period. Our survey suggests the majority felt valued and expressed pride and willingness to participate again should the need arise.

### Conclusion

The rapid establishment and deployment of a dedicated prone positioning team may have contributed to both patient care and staff well-being. We provide insight and lessons that

**Funding:** The authors received no specific funding for this work.

**Competing interests:** The authors have declared that no competing interests exist.

**Abbreviations:** AHRF, Acute hypoxic respiratory failure; ARDS, Acute respiratory distress syndrome; ACCP, Advanced Critical Care Practitioner; COVID-19, Coronavirus Disease 2019; FICM, Faculty of Intensive Care Medicine; GICU, General Intensive Care Unit; ICNARC, Intensive Care National Audit and Research Centre; ICS, Intensive Care Society; ODPs, Operating department practitioners; PPE, Personnel protective equipment; PHE, Public Health England; SARS-CoV-2, Severe acute respiratory syndrome Coronavirus 2; UK, United Kingdom; UHS, University Hospital Southampton.

may be of value for future respiratory pandemics. Future work should explore objective clinical outcomes and long-term sustainability of such services.

## Background

In January 2020, the global pandemic of Coronavirus disease 2019 (COVID-19), caused by severe acute respiratory syndrome coronavirus 2 (SARS-CoV-2) viral infection, posed a significant threat to healthcare in the United Kingdom (UK). It became evident that a subset of COVID-19 patients would experience critical illness, manifesting as acute respiratory distress syndrome (ARDS) and subsequent acute hypoxic respiratory failure (AHRF). The majority of these patients necessitated invasive mechanical ventilation, which has a substantial associated risk of mortality [1]. Initial reports from China and Italy highlighted the strain that COVID-19 placed on critical care resources, with mounting demand for mechanical ventilation and a substantial unmet need for additional qualified intensive care personnel.

Prone position ventilation has long been employed in managing ARDS patients and early evidence suggested potential benefits for COVID-19 patients with severe lung disease [2]. However, the process of proning critically ill, mechanically ventilated patients is time-consuming and labour-intensive. Early in the COVID-19 pandemic, it became evident that existing intensive care staff were unable to perform the required multiple prone cycles without compromising bedside care. Consequently, we determined the need for a dedicated, specialised prone team to address the growing service demands, whilst ensuring patient and staff safety. By late March 2020, in response to the increasing number of patients in the General Intensive Care Unit (GICU) at University Hospital Southampton (UHS), a dedicated critical care prone team was established. UHS is a large teaching hospital with approximately 1200 inpatient beds. GICU is a 32-bed unit. Notably, early Intensive Care National Audit and Research Centre (ICNARC) data indicated UHS GICU as a positive (low mortality) outlier for risk-adjusted 28-day mortality from COVID-19 until August 31st, 2020 [3]. Our group has recently demonstrated that prone positioning in our cohort improved oxygenation indices [4]. We attribute these positive outcomes, at least in part, to the swift implementation of the dedicated prone service.

This report outlines our experience in establishing a critical care prone service at UHS, sharing valuable insights and proposing a blueprint for its rapid reactivation in response to further COVID-19 outbreaks or the emergence of novel respiratory illnesses in the future.

## Methods

### Design and conception

The establishment of a prone positioning team to safely reposition mechanically ventilated patients in GICU required meticulous planning, preparation, and multidisciplinary collaboration. This work was discussed with our research and development department, and as a quality improvement project with participant involvement being purely voluntary, ethical approval was deemed not to be required. We assembled a team of senior clinical leaders and training personnel to oversee the development and implementation. This was completed within a two-week period. Fig 1. outlines the key components of our local process for setting up the service.

Based on staffing requirements, we identified three categories of personnel crucial for successful prone manoeuvres.

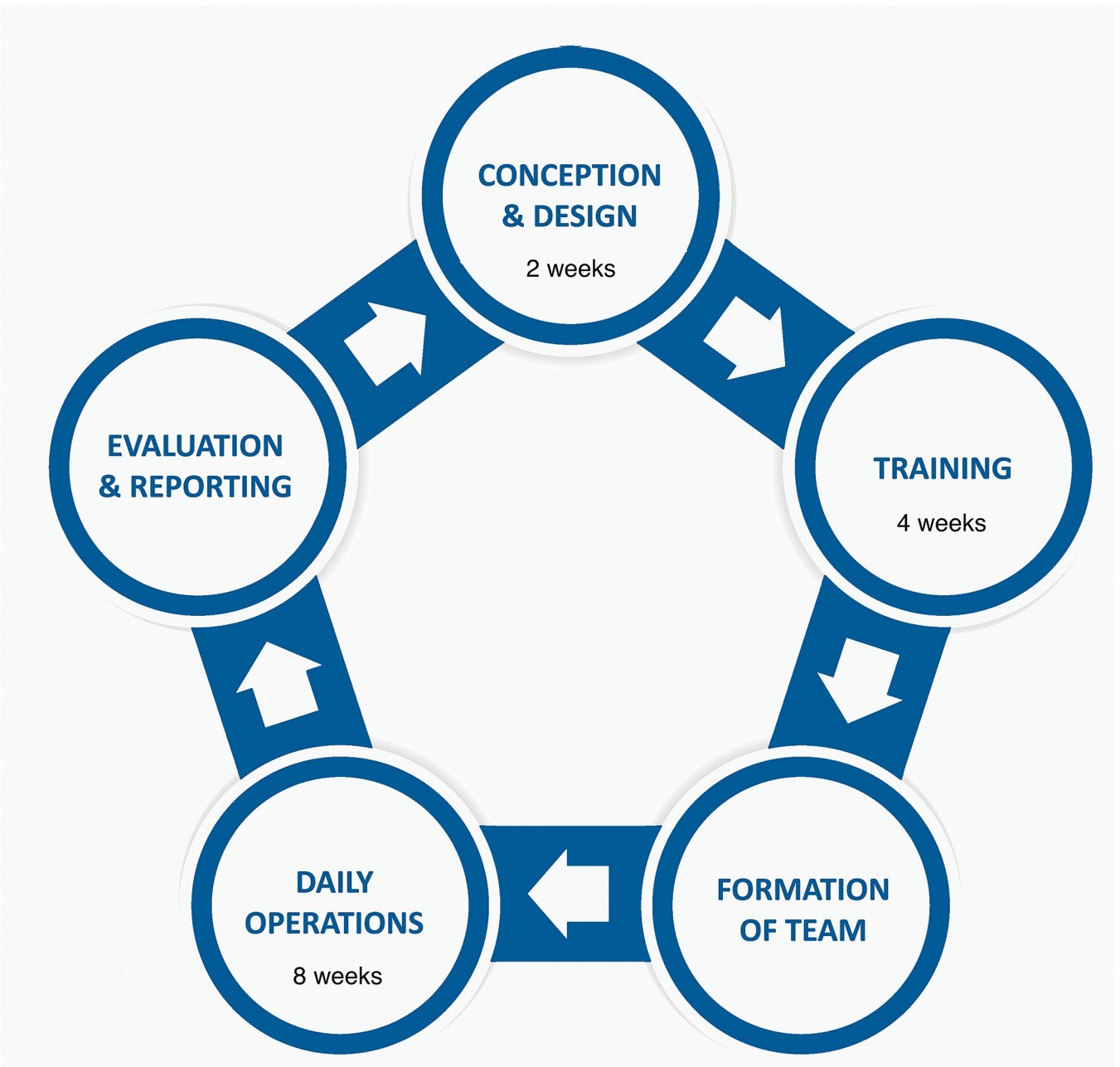

**Fig 1. The stages of development of the prone positioning team during the COVID-19 pandemic.**

1. *Airway-Trained Personnel*
   These clinicians possess the skills to promptly identify and manage airway emergencies that may occur during prone positioning. They are prepared to perform rapid endotracheal intubation if accidental extubation of a critically hypoxic patient occurs. An experienced anaesthetist, intensivist, or advanced critical care practitioner (ACCP) is suited for this role.

2. *Team Members for Patient Turn*
   These individuals possess expertise in manual handling and are responsible for assisting in the physical turning of patients. Operating department practitioners (ODPs), theatre

nurses, and practitioners from relevant surgical specialties (e.g., orthopaedics and neurosurgery) are well-suited for this task.

3. *Team Members for Fine Positioning*
   This role involves precise positioning to prevent complications associated with the prone position, such as pressure injuries or mispositioned lines and circuits. Healthcare workers from the aforementioned specialties commonly manage these issues in theatre settings and are well-equipped to fulfil this responsibility.

## Strategy

**Training and risk mitigation.** Based on simulation exercises, a concise training session of approximately 30–45 minutes was proposed, designed in accordance with the Faculty of Intensive Care Medicine (FICM) and Intensive Care Society (ICS) prone positioning guidelines [5]. Given the potential risks associated with prone positioning for both staff and patients, emphasis was placed on training, practice, and repetition. Staff undergoing training were required to demonstrate satisfactory manual handling competencies. The training sessions encompassed theoretical education, practical demonstrations, and hands-on practice using a high-fidelity setup with an intubated mannequin connected to a ventilator and various infusions and catheters as would be encountered in GICU.

**Resource dissemination.** To facilitate the widespread distribution of information and learning resources across different hospital departments, an online local application was developed (S1 Appendix). This application provided easy access to essential documents such as the FICM and ICS prone position guidelines, training materials (e.g., prone positioning video demonstration), contact information, and a logbook. It was accessible from any browser or mobile device, independent of the staff intranet connection.

**Team assembly.** Approximately 550 staff members from diverse clinical and non-clinical backgrounds received training over a four-week period. All members were existing UHS staff, and no external hiring was required. Recognising the uncertainties surrounding critical care services during the pandemic, a proactive decision was made to significantly scale up the number of staff trained in prone positioning. As knowledge about the virus and clinical outcomes evolved, more targeted training was provided.

**Training delivery and anaesthetic support.** Training sessions were conducted by a dedicated nurse-led skills training team over several weeks, with multiple daily drop-in sessions. Separate sessions were held for approximately 60 anaesthetists, accounting for specific airway, ventilation, and cardiovascular considerations during prone positioning episodes. Simulation exercises guided the plans for ensuring 24-hour availability of experienced anaesthetists within the prone team. Additional tiers were added to the on-call anaesthetic rota to maintain safe anaesthetic coverage for other areas, responding to the increasing demand for anaesthetic support services. Recruitment of outpatient nursing staff and theatre teams further bolstered the team.

**Staffing structure and roster.** Based on training exercises and early experiences with prone positioning in GICU, it was determined that a minimum of five to six members were required in each team. The standard team structure during the peak COVID-19 activity on GICU included one experienced anaesthetist or airway-trained clinician, four team members proficient in manual handling and positioning, and one bedside GICU nurse (typically the nurse assigned to care for that patient). Additional team members were occasionally necessary for repositioning obese or complex patients (such as those with multiple intercostal chest drains etc.).

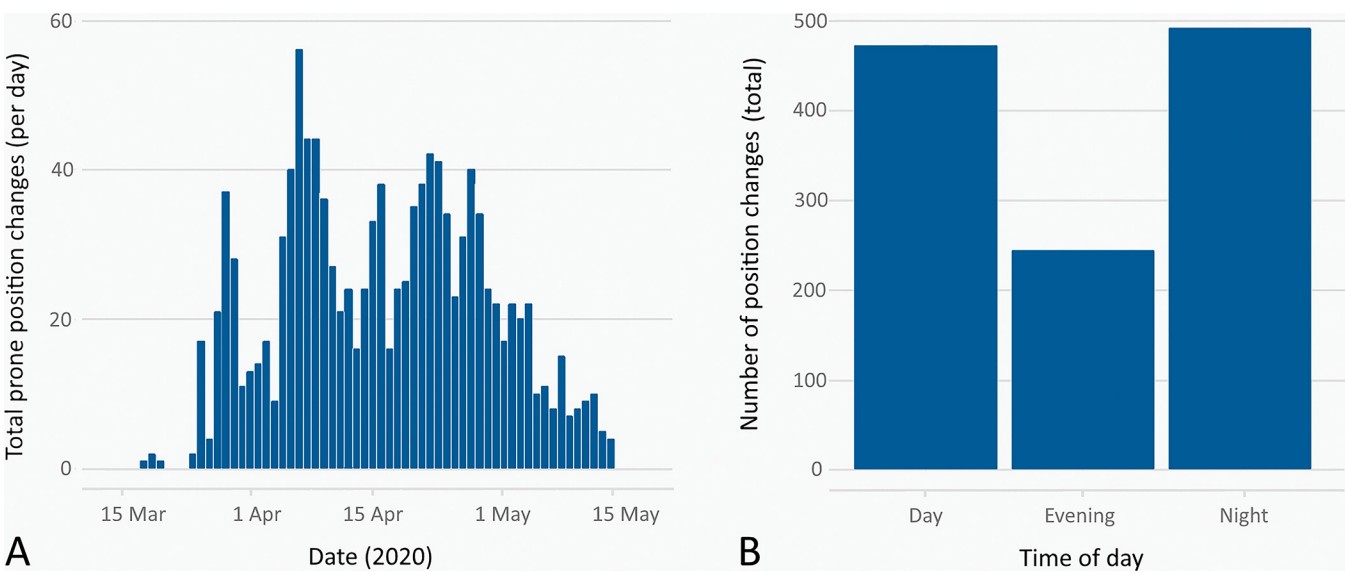

**Fig 2. Prone position changes during peak COVID-19 activity on GICU.** Chart A shows the total number of prone position changes which occurred on a given day, during the peak period of activity. Chart B shows the relationship between total prone position changes and the times of day at which they occurred.

Initially, prone teams were formed on an ad-hoc basis, utilising available anaesthetists supported by nursing, medical, and support staff. They were deployed throughout the four-week training program. Ensuring comprehensive 24/7 coverage with a fully staffed prone positioning team required meticulous organisation, but increasing patient volume necessitated the development of a structured roster to sustainably perform over 40 prone manoeuvres every 24-hour period.

To evaluate the service, we sought regular feedback from the prone positioning team members. Feedback and learning points were disseminated on a regular basis to promote real-time service improvement.

## Results

The prone positioning service at UHS formally commended on March 26, 2020, running continuously for 8 weeks. Utimately, we deployed 78 staff members to full-time prone positioning teams. Minimum staffing for each team was one airway-trained clinician and five other staff members. The same team structure was required for both initial prone positioning, and subsequent re-positions. Throughout the first wave of COVID-19, we mechanically ventilated 184 patients, of which 144 received one or more prone positioning cycles. A total of 1208 prone and repositioning manoeuvres were performed, with a higher frequency of manoeuvres occurring outside regular working hours (Fig 2).

### Key components of a typical prone positioning shift

Fig 3. Provides an illustration of the essential elements of a typical prone positioning shift.

### Importance of team leadership

Based on our experience, designating a team leader was crucial for effective coordination and communication with the GICU medical team. A simple whiteboard system facilitated patient tracking and safe handover. The team leader role was typically fulfilled by a senior nursing team member or an anaesthetist.

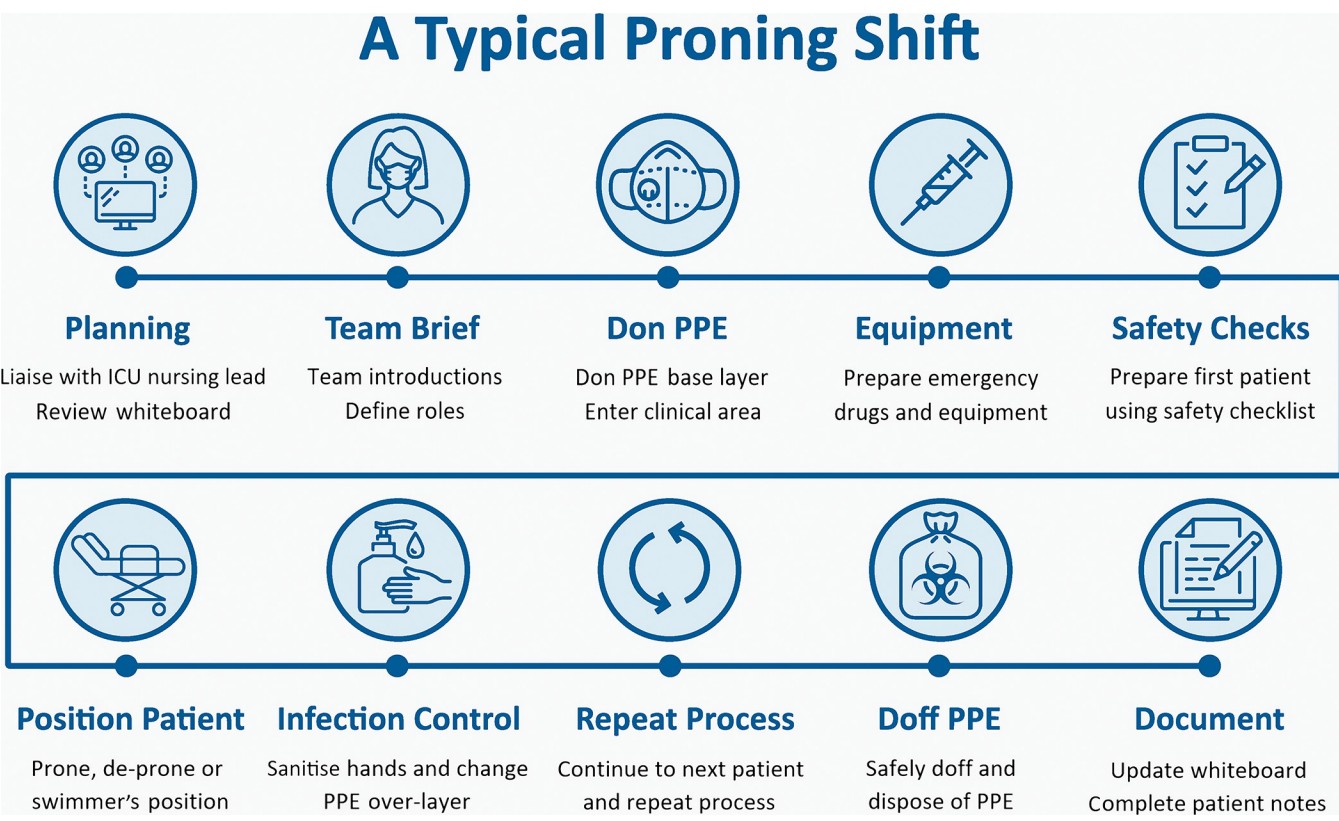

**Fig 3. The key components of a typical prone positioning team shift.** *Whiteboard* refers to a system usually involving a flipchart or wipeable surface used to track patient activity. *PPE* = Personal Protective Equipment. *Donning* refers to the process of applying PPE in a structured manner as directed by local guidance. *The base layer* refers to the primary layer of PPE worn by a clinician. *Over-layer* refers to a disposable, second layer of PPE worn by a clinician when moving between patients in a cohorted area. *Doffing* refers to the process of removing PPE in a structured manner to reduce the risk of cross-infection.

### Team brief and introductions

Given the multi-professional nature of the prone positioning teams and the likelihood of staff members not having worked together previously, we conducted a team brief before each shift. This included introductions with first names, roles, and a brief background experience description of each staff member. The team leader provided relevant information about the patients to be repositioned, including any important clinical details.

### Utilisation of checklists

ICS and FICM guidelines formed the foundation for the locally adapted prone positioning protocols used at UHS. Checklists, encompassing indications, contraindications, and pre- and post-procedural checks ensured standardised and comprehensive practice. Printed and laminated checklists were readily available at each bed space, and electronic versions were accessible via the MicroGuide™ App and the local intranet (S2 Appendix).

### Equipment and drug preparation

To familiarise new prone positioning team members with the critical care environment, a separate equipment and drug trolley was created for easy access to emergency supplies. This approach was derived from simulation-based training sessions and aligned with FICM guidance (4).

### Personnel Protective Equipment (PPE)

Following Public Health England (PHE) guidance [6], enhanced PPE was worn by staff to safeguard both themselves and patients. Considering the finite availability of PPE, we practiced the following; ensured comfort and a secure fit before entering clinical areas, cohorting patients to perform sequential prone positioning manoeuvres during a single PPE session, arranged escorts between GICU overflow areas to minimise PPE doffing, and scheduled prone positioning rounds to avoid coinciding with the clinical team's PPE donning and doffing times.

### Documentation of prone positioning episodes

We established a system to log prone positioning and repositioning events, which was essential for workload estimation, safety event reporting, and service improvement. The online logbook system was implemented, with completion responsibility typically assigned to the lead anaesthetist or team leader on each shift. Unfortunately, we did not explicitly record any adverse events in the logbook, as this would typically be recorded in the clinical notes.

The GICU environment can present challenges for staff without prior exposure, leading to feelings of anxiety. Adequate preparation and support during training and deployment are crucial. Here we highlight the importance of familiarising new team members with the GICU environment, and provide recommendations to improve staff experience and wellbeing during mentally and physically demanding prone positioningshifts.

### Induction and preparation

To address the potential challenges faced by new or inexperienced team members, it is essential to provide comprehensive training. We recommend that new team members initially join the prone positioning team in a supernumerary role, allowing them to become familiar with the intensive care environment and its unique demands.

### Limiting exposure

Working in PPE can be hot and uncomfortable, and prolonged periods in the GICU may increase the risk of cross infection. To improve staff experience and wellbeing, it is advised to limit staff exposure whenever possible. On our unit, in-hour anaesthetic prone positioning shifts were divided between two or three assigned anaesthetists, resulting in positive feedback and improved wellbeing among staff. Additionally, starting staggered prone positioning rounds with sufficient time to complete repositioning manoeuvres, as well as doffing well before handover helps avoid delays and reduces queues at doffing areas.

Continuous auditing and evaluation of service development are essential for improving patient care. While quantifying the impact of individual prone positioning manoeuvres on individual patients is clearly beyond the scope of this report, tracking the overall quantity of manoeuvres performed provides a useful workload metric. We again emphasise the importance of using a logbook here.

In addition to volume metrics, understanding the lived experience of staff members who joined our prone positioning team is crucial. To gather valuable insights on service evaluation, we conducted an internal survey (see supplementary data) targeting 78 full-time team members. The survey comprised 49 questions, and we received 61 responses from a diverse range of healthcare backgrounds.

### Demographic profile

The majority of respondents were female (80%) and aged 25–54 years old (87%). Participants joined the team from various healthcare backgrounds, including medical, nursing, operating theatres, and research teams. 62% had not specifically volunteered for a role.

### Anxiety

Most respondents reported feeling anxious, primarily related to infection risk and concern for spreading the infection to their families. There was also concern about returning to out-of-hours working patterns after working routine hours for some time.

### Training

Formal training on prone positioning was provided to 92% of respondents, with 100% feeling adequately prepared to some extent. 84% had physically practiced on a person or mannequin, and 82% were shown and encouraged how to use our checklist. However, only 78% felt they received sufficient training, and they reported certain resources, such as using the prone positioning team web application, were less clearly emphasised with only 27% reporting they were shown how to use it. Potentially then, the quantity of prone positioning manoeuvres performed was an underestimate.

Of the respondents that had used the web application, 94% found it useful. 39% were aware of our prone positioning team training video, and all found it useful. Free text comments for training reported that some staff were not formally told of training sessions, and happened to find out from colleagues. Some thought that the mannequin size was not reflective of the real patients, and that the available checklists and videos were not highlighted during the face-to-face training sessions. Of the respondents that underwent training, 57% went on the join the prone positioning team, but mentored or supernumerary shifts were available to only 17% of team members. Of those who did not have this opportunity, 70% reported that they would not have found it helpful anyway.

### Team dynamics and roles

Assigned roles varied, with 42% of respondents reporting they had undertaken the role of team leader during a shift. Teamwork within the prone positioning team was highly valued, with 93% reporting good teamwork. When asked specifically if there was good teamwork between the prone positioning team and the resident GICU team the majority (71%) agreed. Some reported mixed messages (%) from GICU teams about which patients required prone positioning, and some raised concerns (%) about perceived tension amongst teams and some obstructive comments about the use of muscle relaxants. There was a particularly strong feeling of teamwork between respondents and anaesthetists, with 98% reporting good teamwork. 95% reported always knowing who the team leader was and 70% reported they would know how to escalate any concern and importantly, felt their concerns would be taken seriously. 34% reported witnessing a clinical problem or emergency during a prone positioning procedure, with the majority (66%) of issues surrounding the ventilator or breathing circuit.

### Checklist utilisation

The majority (93%) reported usually or always using the prone positioning checklist, with 91% finding it useful in preventing patient safety incidents. However, availability of the checklist was a concern, with only 65% reporting it was usually or always physically accessible. 9%

reported that they witnessed incidents that they believed would have been prevented by the use of the checklist.

## PPE and safety

80% reported they had access to appropriate PPE for every shift, but concerningly 62% reported they had spent longer than four hours in PPE at any one time. A small percentage (7%) tested positive for COVID-19, and 7% reported physical injuries. Comments included exacerbation of existing musculoskeletal symptoms, and sore skin from wearing PPE.

## Wellbeing

The reported number of prone positioning shifts undertaken varied, with 19% reporting they completed no shifts, and 2% reporting more than 40. The majority of people (65%) completed between one and 20 shifts. When asked about work intensity, the vast majority reported heavy (65%) or very heavy (7%). Breaks were usually or always sufficient for 84% of respondents, but sadly 11% reported negative psychological health effects, necessitating time off work for recovery.

## Overall satisfaction

Despite the challenges faced, the majority (88%) felt proud to work on the prone positioning team, 82% felt valued, and 90% expressed willingness to join the team again.

## Discussion

Here we present the establishment of a dedicated prone positioning service in response to overwhelming demand during the COVID-19 pandemic. Our aim was to alleviate the work-load and pressure on critical care clinical teams by rapidly training a large number of staff. We discuss the positive impact of safe and timely prone manoeuvres on benchmarked outcomes and staff well-being.

One notable aspect of our approach was the training of a significantly larger number of staff than those who ultimately joined the prone positioning team. This strategy aimed to create a larger pool of trained individuals, which could be considered beneficial. However, it also strained resources. Focusing teaching efforts on the smaller number of staff who joined the team could have further improved their confidence. While the methods we describe are gener-ally transferable, it is important to note that our large, university teaching hospital may have greater access to resources than smaller centres. To gain insight into the lived human experi-ence, we conducted a survey that received a high response rate. Although open surveys are sus-ceptible to bias, the strong participation mitigates some of this risk. Nevertheless, it is worth acknowledging that the survey did not include responses from members of the clinical teams, which may limit our understanding of their perceptions. In addition, we did not record adverse events in our logbook which were typically written in the clinical notes. We would add a reporting feature for any future work.

Our findings align with previous work that emphasises the benefits of a multidisciplinary approach to prone positioning, which reduces complications in critically ill COVID-19 patients [7]. We also address barriers identified in another study, specifically focussing on knowledge and team culture [8]. While other centres have established prone positioning teams, they have not provided detail on training methods or explored the human impact through surveys [9,10]. Some centres with smaller teams limited their service to daytime hours

[11,12], though interestingly their survey results echoed our findings, highlighting general satisfaction but with occasional power struggles.

To ensure the successful implementation or re-initiation of a prone positioning service, several factors should be considered. These include workload projections, the impact on other hospital areas, and collaboration among multiple teams and stakeholders. We recommend the formation of a dedicated "setup" team comprising senior personnel from relevant departments to provide effective leadership. Adequate training, including in-situ, high-fidelity simulation, is crucial, and maintaining a logbook for prone manoeuvres and complications facilitates risk identification, ongoing education, and team development. Prioritising rest and support for team members are essential for maintaining their well-being and performance. Regular feedback from the team members also provides valuable insights for improvement.

Although our work provides valuable insights, some questions remain unanswered. Future work could explore the direct impact of dedicated prone positioning teams on clinical outcomes. Additionally, incorporating the perspectives of clinical team members who were not surveyed would provide a more comprehensive understanding of the service's effectiveness. Evaluating the long-term sustainability and cost-effectiveness of prone positioning services across different healthcare settings is also necessary. Finally, exploring the potential benefits of advanced technologies, such as robotics or artificial intelligence, in prone positioning could open new avenues for future investigation.

## Conclusion

The COVID-19 pandemic was devastating and exhaustive to NHS staff, but has prompted innovative and collaborative approaches to healthcare delivery, yielding substantial improvements in patient outcomes. Here, we share our local experiences of initiating, organising, and implementing a dedicated prone ventilation service. Amid the peak COVID-19 activity in our GICU, we successfully conducted over 1200 individual prone positioning episodes, alleviated the burden on critical care staff and ensured the safe and timely provision of prone ventilation in our patients, contributing to excellent benchmarked outcomes. By sharing our experiences, we hope to inspire and encourage similar initiatives that enhance healthcare delivery and alleviate some of the challenges faced by critical care teams.

## Supporting information

**S1 Appendix. Prone positioning app and logbook.**
(TIFF)

**S2 Appendix. The general intensive care unit prone ventilation checklist.**
(TIFF)

**S1 File.**
(DOCX)

## Author Contributions

**Conceptualization:** Matthew Stubbs, Ahilanandan Dushianthan.

**Data curation:** Matthew Stubbs, Rezaur Rahman, Alexander I. R. Jackson, Helmi C. Burton-Papp.

**Formal analysis:** Alexander I. R. Jackson, Helmi C. Burton-Papp, Ahilanandan Dushianthan.

**Writing – original draft:** Luke Bracegirdle, Matthew Stubbs, Rezaur Rahman.

**Writing – review & editing:** Luke Bracegirdle, Robert Chambers, Sanjay Gupta, Michael P. W. Grocott, Ahilanandan Dushianthan.

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
