## [Decision Letter · Decision Letter 0]

10 Oct 2023

PONE-D-23-22349Organisation and delivery of a dedicated multidisciplinary prone ventilation team in the intensive care unit: strategies and lessons from COVID-19PLOS ONE

Dear Dr. Bracegirdle,

Thank you for submitting your manuscript to PLOS ONE. After careful consideration, we feel that it has merit but does not fully meet PLOS ONE’s publication criteria as it currently stands. Therefore, we invite you to submit a revised version of the manuscript that addresses the points raised during the review process. In particular, the reviewers feel that the aim of your paper is not completely clear, and that the text is sometimes too verbose and not well organized. Moreover, there is a lack relevant outcome measures, both clinical (such as patients survival, eventual increase in ventilator-free days, etc.) and organizational (time to full implementation of your intervention). Generalizability to other hospitals is also an issue, and should be appropriately discussed.

We look forward to receiving your revised manuscript.

Kind regards,

Tommaso Tonetti

Academic Editor

PLOS ONE

Journal Requirements:

2. Please amend your manuscript to include your abstract after the title page.

Reviewers' comments:

Reviewer's Responses to Questions

**Comments to the Author**

1. Is the manuscript technically sound, and do the data support the conclusions?

Reviewer #1: Yes

Reviewer #2: Yes

Reviewer #3: No

Reviewer #4: Yes

2. Has the statistical analysis been performed appropriately and rigorously? 

Reviewer #1: N/A

Reviewer #2: N/A

Reviewer #3: N/A

Reviewer #4: N/A

3. Have the authors made all data underlying the findings in their manuscript fully available?

Reviewer #1: Yes

Reviewer #2: No

Reviewer #3: Yes

Reviewer #4: Yes

4. Is the manuscript presented in an intelligible fashion and written in standard English?

Reviewer #1: Yes

Reviewer #2: Yes

Reviewer #3: Yes

Reviewer #4: Yes

5. Review Comments to the Author

Reviewer #1: The authors of this manuscript aimed to highlight the role of a prone positioning team to alleviate the workload and pressure on critical care clinical teams. They discussed the positive impact of safe and timely prone maneuvers on outcomes. They also pointed out the benefits of a multidisciplinary approach to prone positioning, which reduces complications in critically ill COVID-19 patients.

The manuscript sheds light on an interesting topic, is based on impressive empirical evidence, and makes an original contribution. I only have some minor comments for final improvements:

1- I would recommend adding an abstract that provides a comprehensive synopsis of the contents of the prospective manuscript that consists of 4 paragraphs: Background, Methods, Results, and Conclusions.

2- I have marked a few typos and grammar mistakes which need to be corrected. ( file attached )

3- What was the time elapsed from the conception of the Prone Team initiative to training and first intervention? It would be better to add this information to “figure one” to illustrate the time frame for this stage.

4- What was the final number of the prone team? How many were in each category according to your staffing requirements classification?

5- It would be better to add a table demonstrating the results of the internal survey for the prone team members. And what is the significance of these results?

6- How much was the capacity of the ICUs where the prone team worked at?

7- Can you elaborate on the indications and contraindications of the prone positioning?

8- What were the recorded adverse events that occurred during the prone positioning?

9- A list of all abbreviations used in this manuscript should be added.

Reviewer #2: I read with great interest the study of Bracegirdle and colleagues in which the authors described in details the Organisation and delivery of a dedicated multidisciplinary prone ventilation team in the intensive care unit to face the COVID-19 pandemic. First of all I would like to congratulate everyone on the research team and on the prone ventilation team for working so hard with the aim of improving patient's care. I have few major and minor comments:

Major comments:

At the end of the "Background session" the authors report the UHS GICU as a positive (low mortality) outlier for risk-adjusted 28-day mortality from COVID-19 until August 31st, 2020. The authors state that they attributed these positive outcomes, at least in part, to the swift implementation of the dedicated prone service. Reference 3 is no more available online and should be changed. While it is not my intention to question the beneficial survival effects following team implementation, the statement is not supported by any clinical data and therefore should be removed by the authors. Data regarding mortality would be very interesting but, since the team was established very early in during the first stages of the pandemic, I doubt that comparisons before and after team implementation can be made.

The timeline for a full team rollout is unclear. How many weeks were necessary before the team was fully implemented in the clinical practice?

The authors state that quantifying the impact of individual prone positioning manoeuvres on individual patients would be beyond the scope of the report. I disagree with the authors. Clinical outcomes would be extremely important to fully understand whether such a massive use of resources translates to clear clinical benefits. At least, I would report the incidence of adverse events during manoeuvres before and after the implementation of the dedicated team (incidence of unplanned extubations etc…)

Minor comments:

It is not clear whether manoeuvres of supination following a pronation cycle require there same number of trained personnel. The authors should clarify this aspect.

Figure 1 is not really informative.

It is not clear whether the extra personnel came from other departments or specifically hired to work in the pronation team.

Reviewer #3: In this paper the authors proposed a document that investigate the approach used for prone positioning and described the results obtained in terms of workload and pressure on critical care clinical teams. Furthermore, the evaluate the the positive impact of safe and timely prone manoeuvres on benchmarked outcomes and staff well-being.This paper should be considered as a procedure useful in one dedicated center but I am not sure the results produced could be generalizable in other center. Overall is not clear what is the aim of the paper and how the limited analysis proposed could be useful for other center. The analysis of the results are scanty and in my opinion is limited interest. The use of checklist is interesting but should be verify the internal and external consistency and ithe ncidence of latent error that can impact in terms of the outcome and in terms of the safe of the procedure. Furthermore, should be clarify the impact of the simulation in this specific context.

Reviewer #4: Dear Authors,

Thank you for the work you have done.

The degree of organization certainly influenced the outcome of the patients (as for any catastrophic event).

However I think the work is a little bit prolix and not well organized.

The reader risks getting lost while reading. I tried to suggest some changes (marry some paragraphs, synthesize others).

I believe it could also be useful to build a table to summarize the major outcomes (reduction of adverse events, type of staff trained, staff satisfaction).

6. PLOS authors have the option to publish the peer review history of their article (what does this mean?). If published, this will include your full peer review and any attached files.

Reviewer #1: No

Reviewer #2: No

Reviewer #3: No

Reviewer #4: No

---

## [Author Response · Author response to Decision Letter 0]

14 Nov 2023

Reviewer 1

“I would recommend adding an abstract that provides a comprehensive synopsis of the contents of the prospective manuscript that consists of 4 paragraphs: Background, Methods, Results, and Conclusions”

We agree. We have added a structured abstract into the manuscript. 

“I have marked a few typos and grammar mistakes which need to be corrected”

We agree with all corrections offered and have kept them all.

“What was the time elapsed from the conception of the Prone Team initiative to training and first intervention? It would be better to add this information to “figure one” to illustrate the time frame for this stage.”

We agree this should be clearer. We have updated both the manuscript and Figure 1 to reflect timelines. 

“What was the final number of the prone team? How many were in each category according to your staffing requirements classification?”

We agree this is important information. Total numbers joining the team have been highlighted in the manuscript, and the survey data provides information on demographics including categories. 

“It would be better to add a table demonstrating the results of the internal survey for the prone team members. And what is the significance of these results?”

We considered this point carefully. For initial drafts, our survey data was presented in a table. However, it was the consensus of our group that a narrative summary was more appropriate. For completeness we will upload the full survey data as supplementary material for the interested reader. We do try to highlight the significance of the team members lived human experience, and we hope this is clear from our discussion. 

“How much was the capacity of the ICUs where the prone team worked at?”

We agree this is important contextual information, and have updated the manuscript to reflect this. UHS is a large tertiary hospital with 1200 inpatient beds. We have multiple critical care environments (general, cardiac, neurological, paediatrics) with a total capacity of 54 level 3 beds plus separate HDU capacity. GICU has 32 beds. 

“Can you elaborate on the indications and contraindications of the prone positioning?”

We agree this is important. Indications and contraindications of prone positioning can be found in the section ‘utilisation of checklists’ as well as on the checklist itself (appendix 2).

“What were the recorded adverse events that occurred during the prone positioning?”

Whilst we recorded prone positioning events, we did not explicitly record adverse events in our logbook. Typically, any adverse events would have been documented in the bedside clinical notes. We accept this is a significant limitation of our work and we regret not considering building this in during the planning phase. Any future projects will have adverse event reporting built into our methods. We have updated the ‘documentation of prone positioning episodes’ and discussion sections to reflect this weakness.

“A list of all abbreviations used in this manuscript should be added.”

PLOS ONE does not specify an abbreviation list is necessary. However, we agree that given multiple abbreviations have been used throughout, it seems reasonable to include one. We have added a full list at the beginning of the manuscript.

Reviewer 2 

“At the end of the "Background session" the authors report the UHS GICU as a positive (low mortality) outlier for risk-adjusted 28-day mortality from COVID-19 until August 31st, 2020. The authors state that they attributed these positive outcomes, at least in part, to the swift implementation of the dedicated prone service. Reference 3 is no more available online and should be changed. While it is not my intention to question the beneficial survival effects following team implementation, the statement is not supported by any clinical data and therefore should be removed by the authors. Data regarding mortality would be very interesting but, since the team was established very early in during the first stages of the pandemic, I doubt that comparisons before and after team implementation can be made.”

Thank you for highlighting issues with reference 3. It referred to an ICNARC report published in June 2020, highlighting that UHS was a positive outlier in terms of mortality from COVID-19. I have checked and agree with the reviewer that this report seems to be no longer available. I have therefore updated this reference to include a review article published by our department, that quotes and references the original ICNARC report. We do feel it important to maintain the statement as it provides context for why we feel our model of prone ventilation teams worked so well. 

In regards to clinical data, we actually do have comparison data, but we maintain it is beyond the scope of this manuscript which is designed to be a report of service development. However, we agree that clinical data is important and interesting, so have added a reference to a cohort study published by our group that examines the clinical outcomes of the cohort of patients who underwent prone positioning at our centre. The study highlights that whilst oxygenation indices were improved, the sample size is too small to draw meaningful evidence survival benefit. 

Jackson A, Neyroud F, Barnsley J, Hunter E, Beecham R, Radharetnas M, et al. Prone Positioning in Mechanically Ventilated COVID-19 Patients: Timing of Initiation and Outcomes. J Clin Med. 2023;12: 4226. doi:10.3390/jcm12134226

https://www.ncbi.nlm.nih.gov/pmc/articles/PMC10342481/

“The timeline for a full team rollout is unclear. How many weeks were necessary before the team was fully implemented in the clinical practice?”

We agree this was not clear. As per our response to Reviewer 1, we have updated both the manuscript and Figure 1 to reflect this. 

“The authors state that quantifying the impact of individual prone positioning manoeuvres on individual patients would be beyond the scope of the report. I disagree with the authors. Clinical outcomes would be extremely important to fully understand whether such a massive use of resources translates to clear clinical benefits. At least, I would report the incidence of adverse events during manoeuvres before and after the implementation of the dedicated team (incidence of unplanned extubations etc…)”

We feel the first point about clinical data has been answered in our response above. 

We absolutely agree that the reporting of adverse events is important. As in our response to Reviewer 1, this is a weakness of our work and we regret not recording this. 

“It is not clear whether manoeuvres of supination following a pronation cycle require there same number of trained personnel. The authors should clarify this aspect.”

We agree this is not clear and have updated the manuscript to reflect this. 

“Figure 1 is not really informative.”

We have updated Figure 1 to include timelines. This addresses concerned from both Reviewer 1 and 2, and therefore we suggest keeping it. 

“It is not clear whether the extra personnel came from other departments or specifically hired to work in the pronation team.”

We agree this is not clear and have updated the ‘Team Assembly’ section to reflect this. 

Reviewer 3

“This paper should be considered as a procedure useful in one dedicated center but I am not sure the results produced could be generalizable in other center. Overall is not clear what is the aim of the paper and how the limited analysis proposed could be useful for other center.”

We agree that as a service development report, our results cannot be generalised to other centres. We acknowledge we are a large, well-resourced centre and that smaller centres may not have similar levels of staffing and resources to pull from. We have tried to emphasise this in paragraph 2 of our discussion. 

“The use of checklist is interesting but should be verify the internal and external consistency and ithe ncidence of latent error that can impact in terms of the outcome and in terms of the safe of the procedure.”

We agree that the use of checklists is not infallible, however, the context here is important. This was an exceptional time, when uncertainty and fear of what would come was on everyone’s minds. Arguably, checklists reduced the cognitive workload of what was a very repetitive task performed in difficult circumstances. Our checklist was written with FICM and ICS standards in-mind and we might argue that adherence would only improve safety rather than increase the incidence of error. 

“Furthermore, should be clarify the impact of the simulation in this specific context.”

We agree the impact of training and simulation should be explored. Under “training” in the evaluation section, we highlight that 84% of our team had simulated practice on a person or manikin. In addition, we highlight that 100% felt adequately prepared after our simulation sessions.

Reviewer 4

“However I think the work is a little bit prolix and not well organized.

The reader risks getting lost while reading. I tried to suggest some changes (marry some paragraphs, synthesize others).”

We agree the initial draft required some re-structuring and we hope we have satisfied this. It is now structured as Abstract, Abbreviations, Background, Methods, Results, Discussion and Conclusion.

“I believe it could also be useful to build a table to summarize the major outcomes (reduction of adverse events, type of staff trained, staff satisfaction).”

This article aims to only describe to roll-out of new prone positioning service. We are able to state absolute number team members deployed, and of manoeuvres performed, but exploring outcomes is beyond the scope of this work. As per our response to Reviewer 1 and 2, we have added a reference for a cohort study describing clinical outcomes in our cohort of patients that underwent prone positioning and hope you find it interesting. We have also updated the manuscript to include the significant limitation that our logbook did not record adverse events. 

As per our response to Reviewer 1, we did initially present our survey results in a table, but it was the consensus of our group that a narrative summary flowed much better. We have however now updated the supplementary files to include the full survey responses.

---

## [Decision Letter · Decision Letter 1]

11 Dec 2023

Organisation and delivery of a dedicated multidisciplinary prone ventilation team in the intensive care unit: strategies and lessons from COVID-19

PONE-D-23-22349R1

Dear Dr. Bracegirdle,

We’re pleased to inform you that your manuscript has been judged scientifically suitable for publication and will be formally accepted for publication once it meets all outstanding technical requirements.

Kind regards,

Tommaso Tonetti

Academic Editor

PLOS ONE

Additional Editor Comments (optional):

Reviewers' comments:

Reviewer's Responses to Questions

**Comments to the Author**

1. If the authors have adequately addressed your comments raised in a previous round of review and you feel that this manuscript is now acceptable for publication, you may indicate that here to bypass the “Comments to the Author” section, enter your conflict of interest statement in the “Confidential to Editor” section, and submit your "Accept" recommendation.

Reviewer #1: All comments have been addressed

Reviewer #2: All comments have been addressed

2. Is the manuscript technically sound, and do the data support the conclusions?

Reviewer #1: Yes

Reviewer #2: Yes

3. Has the statistical analysis been performed appropriately and rigorously? 

Reviewer #1: Yes

Reviewer #2: N/A

4. Have the authors made all data underlying the findings in their manuscript fully available?

Reviewer #1: Yes

Reviewer #2: Yes

5. Is the manuscript presented in an intelligible fashion and written in standard English?

Reviewer #1: Yes

Reviewer #2: Yes

6. Review Comments to the Author

Reviewer #1: (No Response)

Reviewer #2: Thanks to the authors for addressing my major concerns. I appreciate that the lack of clinical data and adverse events before and after implementation of the pronation team has been added as a main limitation of the study. I have no additional comments.

7. PLOS authors have the option to publish the peer review history of their article (what does this mean?). If published, this will include your full peer review and any attached files.

Reviewer #1: **Yes: **Ahmed Uosef

Reviewer #2: No

---

## [Editor Report · Acceptance letter]

18 Dec 2023

PONE-D-23-22349R1 

PLOS ONE

Dear Dr. Bracegirdle, 

I'm pleased to inform you that your manuscript has been deemed suitable for publication in PLOS ONE. Congratulations! Your manuscript is now being handed over to our production team.

Kind regards, 

on behalf of

Prof. Tommaso Tonetti 

Academic Editor

PLOS ONE